# Development of a pharmacy-based HIV PrEP service delivery intervention for Washington, District of Columbia (DC): A study protocol

**Waimar Tun[1]***, **Mamaswatsi Kopeka[2]**, **Donaldson F. Conserve[2]**, **Jennifer Gomez-Berrospi[2]**, **Samuel Janson[2]**, **Courtney Johnson[3]**, **Adedotun Ogunbajo[3]**, **Ngozi Joy Idika[2]**, **Jenesis Duran[2]**, **Arianna Lendino[2]**, **Bezawit Bekele[2]**, **Maya Rezende Tsao[2]**, **Sumaiya Nezam[2]**, **Arona Dieng[2]**, **Naana Koranteng-Yorke[2]**, **Bridget Martin[2]**, **BRIDGE Team[4¶]**, **Demarc Hickson[3]**

**1** Population Council, Washington, DC, United States of America, **2** Milken Institute School of Public Health, George Washington University, Washington, DC, United States of America, **3** Us Helping US, People Into Living, Inc., Washington, DC, United States of America, **4** Building Research and Implementation to Drive Growth and Equity (BRIDGE) Research Team, Milken Institute School of Public Health, Washington, DC, United States of America

¶ Membership of the BRIDGE Team is listed in the Acknowledgments.
* wtun@popcouncil.org

**Data Availability Statement:** No datasets were generated or analysed during the current study. All

## Abstract

Pharmacy-based PrEP service delivery models can help address many of the barriers that inhibit the uptake of PrEP. In an increasing number of states, legislation has been passed, or is under consideration, to allow pharmacists to initiate PrEP without a prescription from a physician or other prescriber. However, there is not yet legislation in Washington, DC to allow pharmacy-based PrEP despite its potential to curb new cases of HIV, which disproportionately affect the Black community in the area. The DC Ends HIV Plan has a goal of less than 130 new cases of HIV per year by 2030, which would require that over 13,000 high-risk residents use PrEP. However, in 2021 only 6,724 Washingtonians were taking PrEP. This study seeks to address the absence of critical formative research into the factors that would influence the implementation of pharmacy-based PrEP in Washington DC using the Implementation Mapping (IM) framework. A needs assessment will be conducted through in-depth interviews (IDIs) with pharmacists (n = 6), PrEP providers (n = 6), current PrEP users (n = 6), DC Department of Health officials (n = 2), DC Board of Pharmacy officials (n = 4) and pharmacy-based PrEP experts (n = 4) to provide input on the operational aspects of pharmacy-based PrEP model as a strategy to increase PrEP uptake. Information gathered through this needs assessment will be used to develop standard operating procedures for the introduction of pilot pharmacy-based PrEP into community-based retail pharmacies.

## Introduction

In 2022, there were 11,747 Washington, District of Columbia (DC) residents living with HIV, representing 1.7% of the total population [1]. In 2022, there were 210 newly diagnosed HIV

relevant data from this study will be made available upon study completion.

**Funding:** The proposed research in this publication is funded by the District of Columbia Center for AIDS Research (P30AI117970), which is supported by the following NIH Co-Funding and Participating Institutes and Centers: NIAID, NCI, NICHD, NHLBI, NIDA, NIMH, NIA, NIDDK, NIMHD, NIDCR, NINR, FIC and OAR. DFC was supported by a grant from the National Institute of Mental Health (Grant #R00MH110343: PI: DFC). JGB was supported by the Minority Health International Research Training (MHIRT) grant no. T37-MD001448 from the National Institute on Minority Health and Health Disparities, National Institutes of Health (NIH), Bethesda, MD, USA. The funders had no role in study design, data collection and analysis, decision to publish, or preparation of the manuscript.

**Competing interests:** The authors have declared that no competing interests exist.

cases in DC [1]. Among those who were newly diagnosed in DC between 2017–2021, 1 in 2 were Black men and 1 in 2 were men who have sex with men (MSM) of color [1]. Black men and women account for 71% of people living with HIV in DC, even though they make up just 45% of the population [1, 2]. Given the high HIV burden in DC and the disproportionate impact on Black men and women [3], the Black community of DC is particularly poised to benefit from HIV pre-exposure prophylaxis (PrEP) [4]. PrEP medication has been shown to be highly effective in preventing HIV infection in individuals at high risk of infection [5–7]. However, barriers such as low perception of risk, HIV-related stigma, lack of transportation, and cost can impede access to PrEP services [8–10]. A pharmacy-based approach to PrEP service delivery might help alleviate barriers to accessing PrEP and potentially increase PrEP uptake and adherence, especially among populations with suboptimal PrEP usage but high HIV rates. The DC Department of Health is currently reviewing pharmacy-based PrEP legislative models and is interested in testing alternative PrEP service delivery models [11]. Given this potential legislative change, this is a timely study that aligns with the priorities of government stakeholders and the HIV prevention needs of the population.

We will develop an HIV service delivery model that builds on pharmacy-based approaches (community-based pharmacy and retail pharmacy) for PrEP delivery by using the 6-step Implementation Mapping (IM) process [12]. These steps include conducting a needs assessment, selecting theory-based methods and practical strategies, producing intervention components and materials, crafting an implementation plan, and designing an evaluation plan. Our team will use the Consolidated Framework for Implementation Research (CFIR) [13] to guide the different IM steps. The pharmacy-based PrEP model will be one in which clients can receive PrEP services at the pharmacy; it will be aligned with what is within the DC law regarding PrEP prescribing and dispensing by pharmacists at the time of implementation.

Pharmacy-based PrEP offers an alternative approach to increasing PrEP uptake for individuals unable to access care at traditional facilities, especially those residing in underserved communities and who have marginalized identities. Pharmacies' widespread availability (i.e., 90,000+ pharmacies in the US), regular drug stocks, informal operating hours, familiarity with community residents, and relative anonymity/confidentiality make pharmacies potentially highly preferable sources of care [14–16]. Pharmacists have been vital in public health preventive efforts in the U.S. Since the 1990s, pharmacies have provided immunizations, and in recent years they provided the majority of adult COVID-19 vaccinations [17, 18]. More recently, there is growing evidence for pharmacists to provide PrEP services in the U.S. [16, 19–28]. A systematic review reports early evidence that pharmacy-based PrEP programs are feasible with clients supporting pharmacy access, pharmacists indicating a willingness to offer PrEP, and U.S. clients willing to pay [15]. Various pharmacy-based PrEP models have been implemented in the US. In some models, pharmacists evaluate patients for PrEP initiation, order lab tests, provide education, and prescribe/dispense PrEP under a collaborative drug therapy agreement (CDTA) with a local medical provider [16, 23, 24]. In other models, patients first undergo an initial intake visit at a local healthcare clinic and receive a prescription for PrEP; the pharmacist then handles all follow-up visits including lab assessments, medication dispersals, and any applicable STI treatments [25, 26]. Some models have used telemedicine-delivered PrEP services, whereby patients first perform a pre-visit telemedicine appointment with a healthcare provider, followed by lab test orders that a patient can complete at a local laboratory (e.g., LabCorp). Then, a pharmacist conducts a telemedicine visit to review lab tests, medication, and risk-reduction counseling and sends an e-prescription that can be filled at a local pharmacy of the patient's choosing [27]. These early pharmacy-based PrEP models have shown that HIV screening and other PrEP-related services performed by a

pharmacist are acceptable to patients and that pharmacists can provide PrEP without initial consultation with an HIV medical provider.

Some of the key takeaways from previously implemented pharmacy-based PrEP programs are: i) The need for greater collaboration between pharmacists and providers; ii) The usefulness of HIV testing and sexual health/PrEP counseling by pharmacists; iii) The formation of collaborative practice agreements (CPAs) and CDTAs to expand the scope of practice of pharmacists for PrEP services; iv) The need for appropriate training of pharmacists to provide PrEP counseling, sexual health counseling, and adverse side effect screening; and v) designated space in pharmacies to ensure privacy [24–26]. While early pharmacy-based programs have provided some important lessons, there remain substantial implementation research gaps, particularly with regard to how pharmacy-based PrEP can be implemented given the novelty of the approach. There is a need for a better understanding of what works and does not work, and whether it can be effective in reducing new HIV infections. Given that in 2023, a number of states had proposed and/or passed legislation to expand pharmacy-based PrEP services [29], this work is particularly useful in the context Washington DC, which currently operates through CPAs Currently in DC, although pharmacists do not have legal authority to provide oral PrEP without a prescription from a primary care provider (PCP)/PrEP provider, pharmacists can enter into a CPA with a PCP/PrEP provider [30]. CPAs afford pharmacists greater responsibility for patient assessments, order of related lab tests, drug administration, and initiation, adjustment or discontinuation of medication. If legislation is passed, clients could initiate PrEP at any community pharmacy approved to offer PrEP without the need to establish CPAs with a PCP/PrEP provider. This study will develop and pilot a pharmacy-based PrEP service delivery intervention for Black adults in Washington DC.

## Materials and methods

### Implementation Research Logic Model

The proposed study will consist of formative research to guide the development of a novel pharmacy-based PrEP service delivery model in DC. Fig 1 describes the Implementation Research Logic Model (IRLM) which we will be using for our research; it is guided by key principles and domains from the CFIR [13]. We will conduct formative qualitative research to better understand how the CFIR Determinants outlined in the IRLM affect pharmacy-based PrEP implementation and PrEP uptake. We will use IM to develop the pharmacy-based PrEP service delivery model using the findings from the formative assessment. The Intervention and Implementation Strategies are intended to work through Mechanisms to achieve the proposed Implementation and Client Outcomes.

### Setting

The DC metropolitan area is an ideal location for this research, due to the alarming incidence of HIV among Black adult residents. Nearly 1 in 3 Black MSM in Washington DC are currently living with HIV [1]. Additionally, Prince George's County, Maryland, has the second highest HIV prevalence and incidence in the state of Maryland [31]. While we will not implement pharmacy-based PrEP in Prince George's County, residents of the county could be eligible to utilize pharmacy-based PrEP in DC.

### Study procedures

For the needs assessment (IM Step 1), we will conduct in-depth interviews (IDIs) (Table 1) with pharmacists, PrEP providers, current PrEP users, stakeholders from the DC area (e.g.,

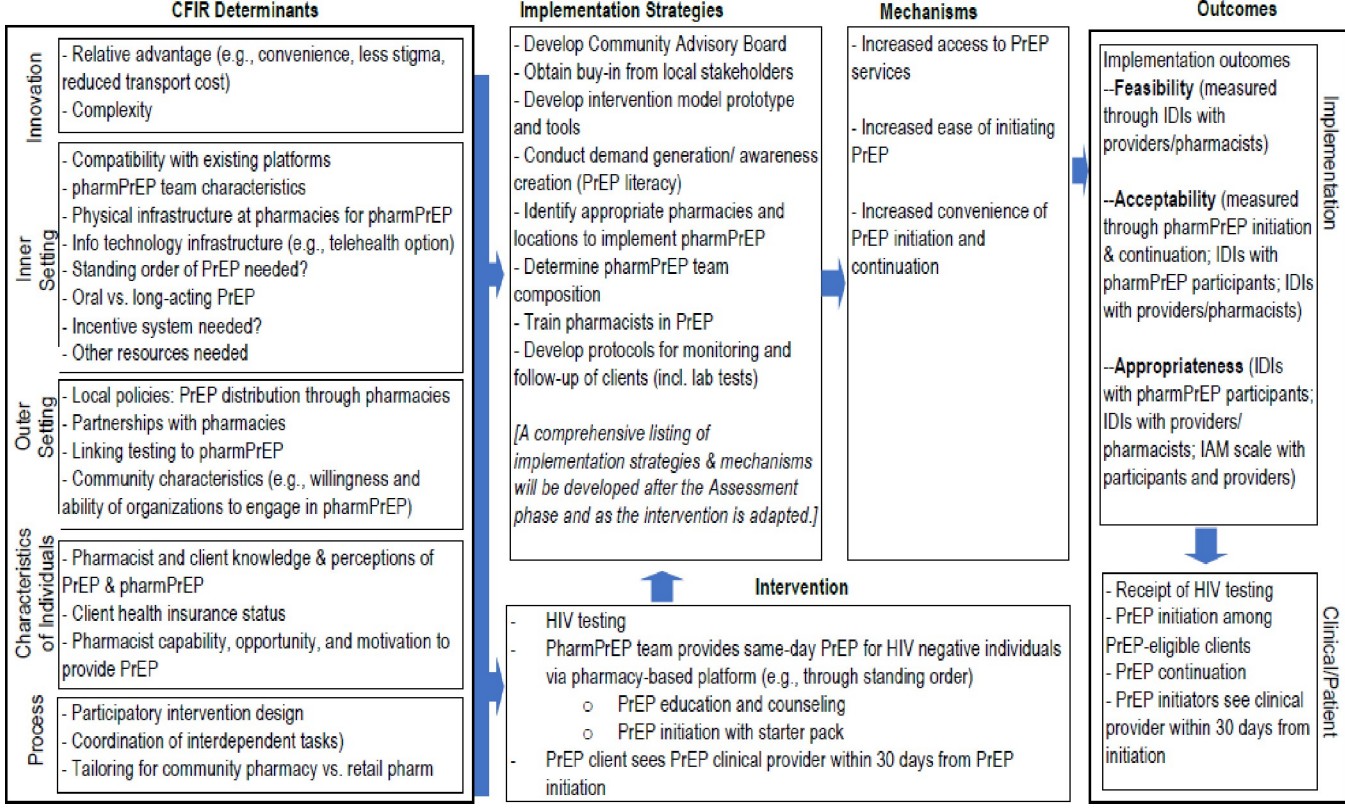

**Fig 1. Implementation Research Logic Model for pharmacy-based PrEP delivery.**

DC DOH representatives, DC Board of Pharmacy), and pharmacy-based PrEP experts to provide input on the operational aspects of pharmacy-based PrEP model as a strategy to increase PrEP uptake (guided by the CFIR).

We are collaborating with a community-based organization (Us Helping Us, People Into Living Inc.) that provides HIV services to the Black community in the DC metropolitan area. The organization currently collaborates with several community/retail pharmacies for its HIV

**Table 1. Formative assessment participants, inclusion criteria, and areas of inquiry.**

| Interview groups | Inclusion criteria | Areas of inquiry & determinants |
|---|---|---|
| **Licensed Pharmacists and Technicians (N = 6)** | Currently working at pharmacy or pharmacy with agreement to provide HIV-related services in DC | Pharmacy-based PrEP model characteristics (e.g., complexity) |
| **PrEP clinical (prescribers) and non-clinical providers (N = 6)** | Provide PrEP to DC residents in the last year<br>Working in DC DoH's Wellness Clinic or other clinics in primarily Black neighborhoods providing PrEP | Inner settings (e.g., pharmacy-based PrEP team characteristics)<br>Outer settings (e.g., policies) |
| **DC Department of Health (N = 2)** | Working as HIV program manager/coordinator with working knowledge of PrEP services | |
| **DC Board of Pharmacy (N = 2)** | Staff of DC Board of Pharmacy | |
| **Pharmacy-based PrEP experts (N = 4)** | Researchers or implementers who have implemented/managed a pharmacy-based PrEP service delivery model in the U.S. | |
| **Current PrEP clients (N = 6)** Black MSM, Black heterosexual women, Black heterosexual men | Black/African American race<br>Age 18 or older<br>Residing in DC area<br>Current PrEP users | Characteristics of individuals and processes (e.g., decision making around pharmacy-based PrEP) |

treatment program. We will purposively recruit pharmacists and pharmacy technicians from these pharmacies, as well as other pharmacies in areas in the city with higher HIV burden. PrEP providers will be selected from local healthcare centers, community-based organizations, and HIV care service providers. The healthcare providers selected will be diverse in affiliation and professional cadre (doctor, nurse practitioner, physician assistant, counselor). Current PrEP users will also be selected from among those currently taking PrEP through Us Helping Us and other community-based clinics; we will diversify based on the longevity of their PrEP use as well as the identities they hold (we are purposefully including Black women, who tend to be underrepresented in PrEP research). Pharmacy-based PrEP experts will be selected from among those who have published/led pharmacy-based PrEP interventions in the U.S., as well as members of or affiliates with the Board of Pharmacy. This study was approved by the George Washington University Institutional Review Board, and has been assigned the reference number NCR234875. Recruitment for the study started on February 10[th], 2024, and is expected to conclude in August 2024. Interested individuals read through and signed an informed consent form prior to the interview.

## Analysis of IDI data

The IDIs will be audio-recorded, transcribed, and two members of the research team will conduct thematic content analysis. Initially, each researcher will read the transcripts independently identifying preliminary codes and subthemes using both inductive and deductive approaches (open coding and coding of theoretical constructs). The focus of the analysis will be to identify how the key determinants identified in the logic model act as facilitators and barriers for pharmacy-based PrEP, with a lens towards tailoring the pharmacy-based PrEP service delivery model for the different sub-groups of Black adults in the DC area.

## Intervention development

Following the analysis of the data collected through the IDIs, the research team will begin the development of the intervention. Some of the key details that will be solidified for the pharmacy-based PrEP model include i) specific responsibilities of the pharmacist and technicians; ii) strategies to facilitate collaboration between pharmacists and PrEP providers; iii) how HIV testing and PrEP eligibility assessments will be conducted; iii) how lab tests will be ordered by the pharmacist or integrated into pharmacy services; iv) who will monitor patients on PrEP (and decide on discontinuation if needed); v) frequency of visit with a physician; vi) what forms of PrEP will be included (pills vs. injection); vii) the role of telemedicine in the intervention.

To work out these details, the remaining steps of the IM process will be utilized, guided by the findings from the IDIs to help identify barriers and facilitators and to define the objectives of the intervention (IM Step 2). We will create matrices of specific intervention objectives at the organizational level (pharmacy level) and community level (community residents), matched with their respective theoretical determinants. The resulting cells of each matrix will contain change objectives that will guide the final selection of theory-based methods and strategies (IM Step 3), and contents and components (IM Step 4) for the pharmacy-based PrEP service delivery model prototype.

The theory selection will be based on preliminary analyses of the IDIs. There is a plethora of theories that could be useful in this study, including the Integrated Behavior Model (IBM), which combines constructs represented in the Theory of Reasoned Action and the Theory of Planned Behavior, to formally analyze the data from IM Step 1 and inform the theory-based methods and strategies for the intervention [29]. IBM posits that a person's intention to

perform a behavior (e.g., seeking pharmacy-based PrEP), is influenced by their attitude (experiential and instrumental) toward the behavior, perceived norm (injunctive and descriptive), and personal agency (self-efficacy and perceived control) [29]. To refine the intervention, we will conduct follow-up consultations with a sample of IDI participants to inform the adoption and implementation plan (IM Step 5) and evaluation plan (IM Step 6).

## Discussion

The proposed study is in line with the DC Ends HIV plan, which includes access to and provision of PrEP as a key priority [31] and can be scalable in a variety of low-resourced, community-based settings (e.g., federally qualified health centers and other HIV service organizations (HSOs)). At a minimum, at the conclusion of this study, we will develop an intervention prototype, including SOPs and intervention tools (e.g., job aides, educational materials. As a next step, we will pilot the developed pharmacy-based PrEP intervention in a small number of pharmacies to examine acceptability, feasibility, and appropriateness. At outset of the pilot phase, participating pharmacists and technicians will complete CDC's Ending the HIV Epidemic Continuing Pharmacy Education Programs (related to HIV testing and PrEP) online training courses [30]. In addition to this training, the PrEP provider and pharmacist at Us Helping Us will provide more hands-on training on the operational aspects of PrEP service delivery. The hands-on training will allow participating pharmacies to work closely with Us Helping Us team who have extensive experience providing community-based PrEP services to the Black population in DC.

If the intervention is found to be acceptable, feasible, and appropriate, we will then develop a proposal for funding to test the efficacy of the pharmacy-based PrEP service delivery model on increasing PrEP uptake and continuation, which will be particularly important as an increasing number of states consider legislations for pharmacy-based PrEP.

A strength of this proposed study is the active collaboration between members of the DC-CFAR (UHU, DC Health, George Washington School of Public Health) and in international NGO (Population Council), who have met regularly to communicate and collaborate on all aspects of the proposed project. We will hold regular project meetings via Zoom to ensure stakeholders are supportive of all aspects of the research and pharmacy-based PrEP implementation. A Community Advisory Board will be formed and consulted before broader dissemination. Findings from this study (after the formative stage and at the end of the pilot) will be shared with DC-CFAR partners, PrEP providers, pharmacists, DC Health staff, HSO stakeholders, and community leaders. We will also share findings at scientific meetings and through peer-reviewed publications.

## Acknowledgments

We would like to thank the members of the BRIDGE research team for their hard work and dedication in this study and in writing the paper: Jennifer Gomez-Berrospi, Ngozi Joy Idika, Jenesis Duran, Arianna Lendino, Bezawit Bekele, Maya Rezende Tsao, Sumaiya Nazem, Arona Dieng, Naana Koranteng, Bridget Martin, Manal Tariq, Suha Ansari, Bukola Rinola, and Samantha Villarreal, who are all at George Washington University's Milken Institute School of Public Health. The lead author for the BRIDGE group for this paper was Jennifer Gomez-Berrospi.

## Author Contributions

**Conceptualization:** Waimar Tun, Donaldson F. Conserve, Samuel Janson, Adedotun Ogunbajo, Demarc Hickson.

**Data curation:** Mamaswatsi Kopeka, Samuel Janson, Adedotun Ogunbajo, Bezawit Bekele, Maya Rezende Tsao, Sumaiya Nezam, Arona Dieng, Naana Koranteng-Yorke, Demarc Hickson.

**Formal analysis:** Mamaswatsi Kopeka, Jennifer Gomez-Berrospi, Ngozi Joy Idika, Arianna Lendino, Maya Rezende Tsao, Sumaiya Nezam, Arona Dieng.

**Funding acquisition:** Waimar Tun, Donaldson F. Conserve, Courtney Johnson, Demarc Hickson.

**Investigation:** Waimar Tun, Mamaswatsi Kopeka, Donaldson F. Conserve, Courtney Johnson, Adedotun Ogunbajo, Arianna Lendino, Bezawit Bekele, Naana Koranteng-Yorke, Demarc Hickson.

**Methodology:** Mamaswatsi Kopeka, Donaldson F. Conserve, Jennifer Gomez-Berrospi, Samuel Janson, Adedotun Ogunbajo, Ngozi Joy Idika, Jenesis Duran, Arianna Lendino, Bezawit Bekele, Naana Koranteng-Yorke, Bridget Martin.

**Project administration:** Mamaswatsi Kopeka, Jennifer Gomez-Berrospi, Adedotun Ogunbajo, Ngozi Joy Idika, Demarc Hickson.

**Resources:** Waimar Tun, Donaldson F. Conserve, Courtney Johnson, Adedotun Ogunbajo, Demarc Hickson.

**Software:** Demarc Hickson.

**Supervision:** Waimar Tun.

**Validation:** Samuel Janson, Courtney Johnson.

**Writing – original draft:** Waimar Tun, Mamaswatsi Kopeka, Donaldson F. Conserve, Jennifer Gomez-Berrospi, Samuel Janson, Courtney Johnson, Adedotun Ogunbajo, Ngozi Joy Idika, Jenesis Duran, Arianna Lendino, Bezawit Bekele, Maya Rezende Tsao, Sumaiya Nezam, Arona Dieng, Naana Koranteng-Yorke, Bridget Martin, Demarc Hickson.

**Writing – review & editing:** Waimar Tun, Mamaswatsi Kopeka, Donaldson F. Conserve, Jennifer Gomez-Berrospi, Samuel Janson, Courtney Johnson, Adedotun Ogunbajo, Ngozi Joy Idika, Jenesis Duran, Arianna Lendino, Bezawit Bekele, Maya Rezende Tsao, Sumaiya Nezam, Arona Dieng, Naana Koranteng-Yorke, Bridget Martin, Demarc Hickson.

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
