## [Decision Letter · Decision Letter 0]

11 Jul 2024

PONE-D-24-22926

Development of a Pharmacy-based HIV PrEP Service Delivery Intervention for Washington, District of Columbia (DC): A Study Protocol

PLOS ONE

Dear Dr. Kopeka,

Thank you for submitting your manuscript to PLOS ONE. After careful consideration, we feel that it has merit but does not fully meet PLOS ONE’s publication criteria as it currently stands. Therefore, we invite you to submit a revised version of the manuscript that addresses the points raised during the review process.

We look forward to receiving your revised manuscript.

Kind regards,

Tinashe Mudzviti, MPhil(MD)

Academic Editor

PLOS ONE

Journal Requirements:

https://journals.plos.org/plosone/article?id=10.1371%2Fjournal.pone.0290631

In your revision ensure you cite all your sources (including your own works), and quote or rephrase any duplicated text outside the methods section. Further consideration is dependent on these concerns being addressed.

3. One of the noted authors is a group or consortium "BRIDGE research team". In addition to naming the author group, please list the individual authors and affiliations within this group in the acknowledgments section of your manuscript. Please also indicate clearly a lead author for this group along with a contact email address.

Reviewers' comments:

Reviewer's Responses to Questions

**Comments to the Author**

1. Does the manuscript provide a valid rationale for the proposed study, with clearly identified and justified research questions?

Reviewer #1: Yes

Reviewer #2: Yes

Reviewer #3: Yes

2. Is the protocol technically sound and planned in a manner that will lead to a meaningful outcome and allow testing the stated hypotheses?

Reviewer #1: Yes

Reviewer #2: Yes

Reviewer #3: Yes

3. Is the methodology feasible and described in sufficient detail to allow the work to be replicable?

Reviewer #1: Yes

Reviewer #2: Yes

Reviewer #3: Yes

4. Have the authors described where all data underlying the findings will be made available when the study is complete?

Reviewer #1: Yes

Reviewer #2: Yes

Reviewer #3: Yes

5. Is the manuscript presented in an intelligible fashion and written in standard English?

Reviewer #1: Yes

Reviewer #2: Yes

Reviewer #3: Yes

6. Review Comments to the Author

You may also provide optional suggestions and comments to authors that they might find helpful in planning their study.

Reviewer #1: The study intends to build evidence within Washington DC of factors that affect pharmacy-based PrEP using the Implementation Mapping (IM) framework. The goal of the intervention is to address the imbalances with regards to access to (unique) services for minorities that are more likely to suffer prejudice from unavailability or inaccessibility of the service. The authors point out that 71% of those affected (and living with HIV) are black men and women thus the intervention will be targeted at this cohort.

It is becoming increasingly true and common that people perceive pharmacists to be easily accessible and trustworthy as evidenced by the uptake of (expanded) services offered in pharmacies with vaccinations against COVID being a recent example where most people were more likely to get the vaccine from their pharmacist than any other healthcare professional. This means the choice of the authors to opt for a pharmacist led intervention is sound.

The authors have demonstrated their knowledge in the subject through the choice of their references which is very impressive and thorough. Thus, I am very impressed by this study protocol and I look forward to the results.

Reviewer #2: The paper is written very well as well as the protocol as presented is good. The paper will guide research in the field in other regions such as those in the global south with much higher HIV prevalence, where the innovation may greatly enhance access to PrEP and/or PEP.

Reviewer #3: This paper proposes a research protocol for the planning, design, implementation and evaluation of PrEP as delivered via pharmacies in Washington, DC. Noting that DC lacks enabling policy legislation to allow for such a service, the paper uses an implementation mapping framework to address gaps in the current knowledge of PrEP services in the area, starting with a modest needs assessment of key stakeholders, including 6 pharmacists, 6 current PrEP providers, 6 current PrEP users, 2 public health officials, 4 pharmacy officials, and 4 experts in pharmacy-delivered PrEP. The complete process will include, in addition to a needs assessment, theory-based methods and practical strategies, intervention components and materials, an implementation plan, and an evaluation plan. The next step is described in the last sentence in the abstract: ‘Information gathered through this needs assessment will be used to develop standard operating procedures for the introduction of pharmacy-based PrEP into community-based retail pharmacies.’

This is a timely topic of some urgency, given the ongoing transmission of HIV among vulnerable populations such as MSM, TGW, and women of color in DC. The authors are to be congratulated for a thoughtful approach to their needs assessment, and for assembling a credible list of investigators.

Introduction

Pg 3-5: The authors make a compelling case here and in the last paragraph on page 6 for the need for expanding access to PrEP in Washington DC and the known and potential benefits of pharmacy-based services.

Pg 4, para 1: The authors need to elaborate on the last sentence that the emerging plan will conform to local DC law regarding prescribing and dispensing PrEP by pharmacists. What would be allowed under current law, and what options might be available if pending enabling legislation is passed? Is there also an option to continue with an implementation research study in the absence of such legislation?

Pg 5, para 1: Do pharmacies in DC currently offer HIV testing? Additional discussion on the implementation and training of staff around HIV testing is needed in the introduction and/or the discussion.

Page 5, para 2: Regarding the training needs, have the investigators reached out to the local HIV training authorities as collaborators, e.g. the regional AETC and the local health department? A bit more consideration of the practical elements in training to scale up this program is warranted.

Pg 6, first sentence: The authors should clarify the rationale for targeting a single vulnerable group, i.e. black adults, and the approach and applicability of the service implementation plan for other at-risk individuals. If an inclusive approach based on HIV risk is to be pursued, as well as a focus on reaching black adults in DC, that should be stated.

Pg 6, first sentence: The reference relates to telehealth in Iowa. Is there specific pending legislation in D.C. that the authors can allude to?

Methods

Pg 15 – The partnership with Us Helping Us is well-conceived and described. This element in the design of the NA is critical. The specific inclusion of black women is also essential.

Pg 8 – Intervention development: The authors should address how they will assess capacity for the PrEP work among the pharmacy interviewees and in the affected communities. Will there be an effort to generate public and/or private funding from this effort? On page 9, the authors indicate an interest in developing a research program with R01 funding to support the test of the efficacy of the service that is developed. What alternatives would be available if the R01 were not funded?

The iterative generation of plans and then a second review by the planning group is important for refinement and modification of the plan.

Results: The plan for a needs assessment and the overall plan to develop pharmacy-based PrEP are offered as results for this paper.

Discussion

The authors expect, ‘at a minimum’, that the study will lead to an intervention prototype and tools, technical and educational products, and further scientific papers on the method and outcome. The authors would be well advised to focus on the substance of the paper, i.e. the needs assessment and its strengths and limitations, as well as the key next steps for deploying PrEP in pharmacies in black communities in DC, rather than offering promises of future actions that may or may not take place.

For example, the authors could use their literature review of pharmacy-based PrEP to consider likely elements of the plan and its implementation in the discussion without limiting the options that will be pursued based on the needs assessment. Do the authors imagine that each pharmacy will have a local PharmD ‘leader’ to manage the start-up and engage people at risk of HIV? Will a community liaison collaborate with the whole process, or perhaps with each participating pharmacy? If community ‘accompaneurs’ are envisioned, how will they be recruited and trained? Will they be all volunteer, or will the group seek to offer reimbursement for this key service?

7. PLOS authors have the option to publish the peer review history of their article (what does this mean?). If published, this will include your full peer review and any attached files.

Reviewer #1: **Yes: **Luckmore Bunu

Reviewer #2: **Yes: **Dr Amos Marume

Reviewer #3: **Yes: **Renslow Sherer MD

---

## [Author Response · Author response to Decision Letter 0]

31 Aug 2024

Thank you for taking the time to read through our manuscript, and to give constructive feedback to our team. Please find our detailed and specific responses attached to this submission.

---

## [Editor Report · Decision Letter 1]

24 Sep 2024

Development of a Pharmacy-based HIV PrEP Service Delivery Intervention for Washington, District of Columbia (DC): A Study Protocol

PONE-D-24-22926R1

Dear Dr. Kopeka,

We’re pleased to inform you that your manuscript has been judged scientifically suitable for publication and will be formally accepted for publication once it meets all outstanding technical requirements.

Kind regards,

Tinashe Mudzviti, MPhil(MD)

Academic Editor

PLOS ONE
---

## [Editor Report · Acceptance letter]

7 Oct 2024

PONE-D-24-22926R1 

PLOS ONE

Dear Dr. Kopeka, 

I'm pleased to inform you that your manuscript has been deemed suitable for publication in PLOS ONE. Congratulations! Your manuscript is now being handed over to our production team.

Kind regards, 

on behalf of

Dr. Tinashe Mudzviti 

Academic Editor

PLOS ONE